# The Effects of Hot-Pack Coating Materials on the Pack Rolling Process and Microstructural Characteristics during Ti-46Al-8Nb Sheet Fabrication

**DOI:** 10.3390/ma13030762

**Published:** 2020-02-07

**Authors:** Haohong Huang, Minle Liao, Qikai Yu, Guohuai Liu, Zhaodong Wang

**Affiliations:** 1School of Materials Science and Engineering, Northeastern University, Shenyang 110819, China; 2State Key Laboratory of Rolling and Automation, Northeastern University, Shenyang 110819, China; zhdwang@mail.neu.edu.cn

**Keywords:** TiAl alloy, pack rolling, sheet fabrication, interface protection, hot-deformed microstructure

## Abstract

The effects of the package materials on the hot workability and stress-strain characteristics of high-Nb TiAl alloy with a nominal composition of Ti-46Al-8Nb (in at.%) were systematically studied via “sandwich structure” hot compression. TiAl sheet fabrication was conducted by hot pack rolling, and the microstructural characteristics and deformation mechanisms were investigated. Based on the analysis of compressed samples and stress-strain curves, the stainless steel/TiAl structure showed better deformation compatibility with homogeneous deformation and decreasing resistance. However, severe interfacial reactions were inevitable. Meanwhile, for the titanium alloy/TiAl structure, few interfacial reactions happened, but wavy deformation and high resistance complicated the compression process. Finally, a package structure with an outer stainless steel isolation layer and inner titanium alloy was determined for the pack rolling process. A TiAl sheet with no crack defects was obtained with 80% reduction. The pack-rolled TiAl sheet took on alternate microstructure of the grain-boundary Al-enriched ribbons and elongated lamellar colonies ribbons. The grain-boundary recrystallized α2 phase, lumpy γ phase, and massive α2/γ lamellae could be observed, which led to the scatter microstructure. The microstructural characteristics mainly resulted from the solute segregations of as-cast Ti-46Al-8Nb alloys, which triggered the local flow softening and deformation incompatibility during hot pack rolling.

## 1. Introduction

TiAl-based alloys have attracted much attention as potential candidates for high temperature structural applications due to their excellent low density, high-temperature properties and creep resistance [1,2]. TiAl sheets can be supplied for the skins, nozzles, tiles, and thermal protection systems (TPS) of aerospace vehicles and have a wide range of applications [3,4,5]. However, large-scale TiAl sheet fabrication has been limited by its poor ductility, low workability, and high-temperature resistance [5,6]. Now, advanced rolling facilities and technologies have been employed for intermetallic TiAl sheet fabrication, such as the isothermal rolling [7,8], direct casting [9], and hot pack rolling [10], among which hot pack rolling is considered the most low-cost and widespread method. During hot pack rolling, the packaging materials can be used for preventing heat loss, surface oxidation, and crack initiation, which makes it possible to produce TiAl sheets on a conventional mill, and the only successful production of large-scale sheets (1800 × 500 × 1 mm^3^) was achieved by GKSS and Plansee (company) with the advanced plate rolling process (ASRP).

Nevertheless, the pack rolling process is not feasible enough for a sound TiAl sheet, especially for the thin-plate fabrication of the TiAl ingot with the intense heat-loss and strain sensitivity. In general, the pack rolling mainly depends on the package structure’s design and the materials, which directly determine the stress-stain conditions, heat protection, and sheet surface quality. The “black box” mode for the package structure always confuses the design for the pack rolling process [11]. Semiatin et al. have indicated the effect of the packaging materials on TiAl sheet fabrication, in which the packaging materials with low flow stress impose tensile stress on the TiAl matrix, while those with high flow stress lead to a decreased temperature and increased strain, causing crack propagation [12]. Although plenty of the studies have been done regarding microstructural evolution and rolling technologies, the crack defects and interfacial reactions still exist, such as edge cracks, heat cracks and Fe–Ti reactions. Moreover, the effect of the package structure on the TiAl sheets under different deformation conditions has not been deeply investigated [13,14]. Therefore, the investigation of the pack rolling conditions for the high-quality TiAl sheet fabrication is significant for its widespread application.

Additionally, the softening behavior and heat-deformed microstructure of TiAl alloy during the hot pack rolling process are closely connected with the original microstructure and rolling conditions [15,16]. For high-NbTiAl alloy, the inhomogeneous microstructure and solute segregation (S, β, and α segregation) have been indicated in as-cast TiAl ingots [17,18]. Simultaneously, the solute addition has significant effects on stacking fault energy of TiAl alloy, which may promote complex interactions between local flow and the recrystallization process during pack rolling process [19,20,21,22]. So far, little attention has been focused on obtaining the fine and homogeneous microstructures of high-NbTiAl alloys by optimizing the package configuration and deformation parameters during hot pack rolling.

In this study, the pack rolling process and microstructural characteristics for Ti-46Al-8Nb sheet fabrication were investigated. Firstly, different packaging materials (0Cr25Ni20 stainless steel and Ti-6Al-4V alloy) were selected for the “sandwich structure” hot compression experiments. Then, the deformation compatibility, stress-strain conditions, and surface reaction were studied between the packaging layer and TiAl matrix. The optimal package structure design for Ti-46Al-8Nb sheet fabrication is presented. Finally, the microstructural characteristics and flow behavior were investigated for the pack-rolled Ti-46Al-8Nb alloy.

## 2. Materials and Methods

The as-cast cylindrical Ti-46Al-8Nb ingot with an actual composition of Ti-46.3Al-7.8Nb (at.%) was fabricated by induction skull melting (ISM). Chemical analysis results were analyzed by inductively coupled plasma mass spectrometry (ICPMS) for both the alloys. Subsequently the homogenization heat treatment was performed at 1200 °C for 4 h. The microstructure is comprised of the α2/γ lamellar colony, B2 phase, and grain-boundary γ phase [23]. The samples were machined from the ingot by a spark machining and sandwiched with the packaging layer for the hot compression by the plane strain technique. The 10 × 10 × 20 mm piece of TiAl alloy and the 4 × 10 × 20 mm piece of the packaging material were designed for the hot compression experiment, during which the stainless steel (2520, Fe-25Cr-20Ni) and titanium alloy (TC4, Ti-6Al-4V) were selected as the packaging materials to investigate their effects on the workability of the TiAl alloy. In a simplified look at rolling conditions, there is compressive stress along the normal direction and tensile stress in the rolling direction, and material flow takes place in the rolling direction; i.e., plane strain conditions [24,25]. For the selected sandwich structure, the metal plate is placed between two indenters where the material flow is promoted along its width and neglected along the length in the plane strain compression. The hot compression above can reproduce the rolling conditions to obtain the plentiful useful flow data for pack-rolled TiAl alloys. The compression samples were heated to 1250 °C at a rate of 10 °C/s, and then a wide range of reductions (10%–60%) were conducted at the strain rate of 1 s^−1^. Three trials were performed for each reduction to reduce errors. The temperature was measured by W/Re thermocouples in the TiAl matrix material.

The packaging samples for Ti-46Al-8Nb alloy are also prepared from the as-cast ingot by ISM after the homogenized heat treatment at 1200 °C for 4 h, and the hot-rolling process was conducted on a mill with a roller 120 mm in diameter and 300 mm in width, which was preheated to 1250 °C for 30 min before rolling. Each sample was rolled at a speed of 40–50 mm/s with a reduction of 8% per pass, and the re-heating was conducted between each rolling pass. After the hot compression and the package rolling experiments, the specimens were cut for microstructural analysis performed by optical microscopy (OM, Axio Lab. A1 Mat, Carl Zeiss AG, Oberkochen, Germany) and scanning electron microscopy (SEM, SIGMA 300, Carl Zeiss AG, Oberkochen, Germany) with an energy dispersive spectrometer (EDS, Oxford, PANalytical, Netherlands). Samples for OM and SEM observations were rubbed by #100, #400, #1000, and #1500 sandpaper, mechanically polished by a Cr_2_O_3_ polishing solution, and etched with a solution of 10 mL HF + 10 mL HNO_3_ + 180 mL H_2_O. The different phases present in the microstructure were discriminated based on their contrast on back-scattered electron (SEM, SIGMA 300, Carl Zeiss AG, Oberkochen, Germany) images, and the component distribution was determined by EDS. Specimens for transmission electron microscope (TEM, FEI, Hillsboro, OR, USA) were prepared by twin jet polishing. This was carried out in a solution consisting of 5% perchloric acid, 35% butanol, and 60% methanol (in volume fraction) at −25 °C at a voltage of 25–30 V.

## 3. Results and Discussions

### 3.1. Deformation Conditions during the Pack Rolling Process

Figure 1 shows the macro-images of the sandwich samples with the different deformation degrees after the hot compression experiments of the Ti-46Al-8Nb alloy. A severe reduction of the packaging layer for both the stainless steel and TC4 alloy layers was obtained due to their low high-temperature strengths. Filmy TC4 packages can be observed at the high deformation degree, showing the layer softer than stainless steel (in Figure 1). Additionally, the unstable deformation of the TiAl layer can be observed with the stainless steel package at the deformation of 40%–60%, which is mainly a result of the wavy contact interface, as shown in Figure 1a. Generally it was seen that the packaging materials with different strengths directly determined the heat-deformed conditions of TiAl alloy, which always plays the function of preventing the heat dissipation, oxidation, and crack initiation during the pack rolling process. The detailed analysis of the deformation conditions with the different packaging materials will be presented later.

The respective deformations of the packaging layers and TiAl matrix for the hot-compressed sandwich samples are shown in Figure 2. Both the packaging layer and TiAl matrix deform with the increase of the total deformation, during which the TC4 alloy layer always takes on the severe reduction while the stainless steel changes laggardly compared to the TiAl matrix, as shown in Figure 2. For the stainless steel package, the increasing reduction is consistent with that of the TiAl matrix, which is smooth with the increasing total deformation of the sandwich samples, as shown in Figure 2a. For the TC4 alloy package, the discontinuous reduction of the packaging layer and TiAl matrix can be observed with the increasing total deformation, during which the sharp deformation of TC4 layer is first observed at the initial period, and it then shifts to the TiAl matrix corresponding to the deformation limit of the TC4 alloy at the total reduction of 30%, as shown the shadow region in Figure 2b. Finally, the wavy reductions of the TC4 package and TiAl matrix are observed with the increasing total deformation during the hot compression process. Through the outcomes above, the selected packaging layer has a great effect on the deformation homogeneity and strain conditions of the TiAl matrix; the TC4 alloy, with lower strength, always leads to heavy reduction and stress concentration during the rolling process, and the packaging materials with the approximate strength of the TiAl alloy should be proper for the pack-rolled TiAl alloy.

Furthermore, the calculated stress-strain curves were obtained for the different packaging materials and deformation degrees during the compression process in Figure 3. The stress values increased continually to the peak stress with the increasing strain. For the stainless steel package, the stress values increased to the peak stress quickly with a small reduction, which shows the homogeneous deformation with the TiAl alloy. Meanwhile, for the TC4 alloy package, the stress values increased slowly with the increasing strain at the initial stage, showing the stress-strain characteristic of the TC4 alloy, and then the peak stress was observed at the strain of nearly 40%. Simultaneously, it is noteworthy that the final peak stress with the TC4 package is always higher than that with the stainless steel package at a high reduction level (≥50%), which can be aroused by stress concentration and heat dispersion. Those were the cause of reduction for the TC4 alloy package, as shown in Figure 3. Generally, the stainless steel package with the high-temperature strength can supply the deformation homogeneity, low-resistance, and heat conservation during the hot deformation process, which should be proper for resolving the crack defect for TiAl sheet fabrication.

Additionally, the high chemical affinity of the TiAl alloy should be always considered during the high-temperature conditions. Figure 4 shows the interface morphology and solute distribution at the contact interface between the packaging layer and TiAl matrix. For the stainless steel layer, a severe interfacial reaction can be observed with the width of nearly 20 μm, and the partial melt of the contact interface was observed due to the local reaction heat; the directional dendritic crystals were vertical to the reaction layer in the TiAl direction, as shown in Figure 4a,b. Additionally the main solutes of Fe, C, Ti, and Al were nearly mixed and the rough interface existed along the reaction layer range (Figure 4c1–c4). The severe interfacial reaction above could be due to the solute discrepancies and high chemical affinity of TiAl alloy. Fortunately, it was observed that the reaction layer was limited and straight along the contact interface for theTC4 alloy package, and the solute Ti and Al elements were separated strictly along the contact interface between the TiAl and TC4 alloy layer (Figure 4e1,e2), which shows that the titanium alloy package can effectively prevent the interfacial reaction during the hot deformation process, as shown in Figure 4d.

Through the results above, the selected packaging materials are critical to the hot deformation processing for the TiAl workpiece, which should own the characteristics of deformation compatibility, heat protection, and low interfacial reactivity. For the stainless steel package, excellent deformation compatibility and heat protection can be obtained owning to its high-temperature strength, but the serious interfacial reactivity may lead to inhomogeneous deformation and crack defects due to the eutectic reaction of Ti-Fe interface at the temperatures of nearly 1100 °C. For TC4 alloy package, the interfacial reaction can be neglected, but the wavy large-reduction and the heat loss always occur due to its low strength during the hot deformation process. Therefore, the deformation compatibility and interfacial reaction should be considered for the TiAl sheet fabrication during the package rolling process, which is mainly centered on the design of the packing materials and packing structure.

### 3.2. Pack-Rolling and the Microstructure of Ti-46Al-8Nb Alloy

Based on the analysis above, the complex design with the outer stainless steel/isolation layer and inner TC4 alloy/coating structure was selected for the pack-rolled Ti-46Al-8Nb alloy, and the sheet macrograph, interface conditions, and macrostructure are shown in Figure 5. The sound TiAl sheet with 140 × 50 × 1.5 mm dimensions can be observed with 85% reduction under the rolling conditions, and the intact surface with no crack defects is obtained through removing the package’s covering and polishing the oxidation skin, showing the proper packing structure’s design and the rolling conditions, as shown in Figure 5a. The packaging layer, isolation layer, and TiAl matrix are clearly distinguished with the straight interface along the rolling direction, as shown in Figure 5b. The thin diffusion interface is obtained between TC4 layer and TiAl matrix, and also between the TC4 layer and the outer stainless steel, as shown in Figure 5c, which shows the favorable interface protection through the packaging materials and structure design for the sheet fabrication. Finally, the macrostructure of the high-deformed TiAl sheet takes on the alternate structure of the dark grain-boundary Al-enriched ribbons [23] and the elongated lamellar structure, as shown in Figure 5d. The detailed microstructural evolution and flow behavior for the heat-deformed Ti-46Al-8Nb alloy is presented later.

Furthermore, the heat-deformed microstructure of the pack-rolled Ti-46Al-8Nb alloy was investigated with increasing degrees of deformation, as shown in Figure 6. The lamellar colonies first take on a nearly 45° direction to the rolling direction with the small deformation degree (30%–40%), and then deflect gradually parallel to the rolling direction with increasing deformation degree, as shown in Figure 6a–c. Additionally, the elongated lamellar colony and dark grain-boundary Al-enriched ribbons are observed during the rolling process, in which the local recrystallization first occurs at the grain boundary with 40% deformation. Then, the recrystallization region gradually increases, accompanied by the breakdown of α2/γ lamellar colony; and finally, the massive recrystallization and elongated retained α2/γ lamellar [26,27] are obtained in the pack-rolled Ti-46Al-8Nb alloy, as shown in Figure 6d.

The results above are mainly attributed to the anisotropic plastic flow of lamellar colonies during rolling process. According to the calculation of polysynthetically twinned crystal (PST) TiAl at 1100 °C [28], the yield stress of lamellar colonies depends on the lamellar orientations between lamellar interface and loading axis, which would be high when the loading axis is perpendicular (about 260 MPa) or parallel (about 180 MPa) to the lamellar interface, while it would be lowest (about 70 MPa) when the lamellar orientations are intermediate, which are easily deformed and of soft orientation. During the hot pack rolling process, the lamellar colonies are sheared, kinked, and rotated first. Under the shear and vertical stress conditions during the rolling process, the lamellae with intermediate orientations may be adjusted to hard orientations (be vertical or parallel to the stress direction). Plenty of lamellar colonies then tilt to the rolling direction. Finally, parallel lamellae are obtained mainly due to their longitudinal extension. Moreover, the stress concentration at the colony boundaries would be easily generated by the anisotropic plastic flow among the lamellar colonies. Local recrystallization and flow softening occur, which are accompanied with the transition of α2/γ lamellar to the α and γ-equiaxed grains process under the temperature and stress conditions. In the case of the studied Ti-46Al-8Nb alloy, the severe Al segregation and massive recrystallized γ phase with low stacking fault energy could accelerate the local flow process at grain boundaries, while the plastic flow of the lamellar colonies is hardly activated due to their high deformation resistance. The deformed microstructure of the local dynamic recrystallized (DRX) grains and elongated retained-lamellar colonies would be eventually obtained in pack-rolled Ti-46Al-8Nb alloy.

Through the results above, the local flow softening exists in heat-deformed TiAl alloy, which may lead to complex microstructure under the pack rolling conditions, especially for the selected Ti-46Al-8Nb alloy with the highly solute segregation. Generally, the streamline structure, multiple- recrystallized grains, and retained lamellar with rolling reduction of 80% are observed in pack-rolled Ti-46Al-8Nb alloy, as shown in Figure 7. The local flow softening mentioned above leads to the inhomogeneous deformation between the grain boundary and the lamellar structure with the recrystallization and phase transition process, which may induce the streamline structure with the alternation of the recrystallized region and the elongated lamellar structure, as shown in Figure 7a. Based on the local recrystallization process, the globalized γ and α phase can grow first at the grain boundary, during which the lumpy recrystallized γ grains are always obtained due to the Al enrichment and the high temperature conditions, as shown in Figure 7b,c. Simultaneously, plenty of twinning of varying thickness is observed in the γ phase under the rolling conditions, as shown in Figure 7c, and it may be related to the very low stack fault energy of the alloy. The more operative slip systems in γ-TiAl can be easily activated during high strain energy deformation at elevated temperature, and the deformation of twins and dislocations further promote the nucleation of DRX γ grains [29,30].

Additionally, the partial lamellar microstructures cannot be broken down completely by hot deformation processing, especially for the Ti-46Al-8Nb alloy with a local flow softening process, and massive retained lamellar colonies with nearly 60 μm are obtained in the pack-rolled TiAl sheet, as shown in Figure 7d, which easily give rise to the scatter mechanical properties of the TiAl sheet. Meanwhile, the original lamellar colonies can be decomposed to the mixed structure of the fine lamellar and recrystallized γ phase under the high deformation conditions, which always has the required characteristics, with its fine and homogeneous microstructure, as shown in Figure 7e. TEM images show that the microstructure above is mainly from the decomposition of α2/γ lamellar structure to the equiaxial α and γ grains, as shown in Figure 7f.

Through the results above, the typical deformed-ribbons structure, lumpy γ phase, and retained lamellar structure are always obtained in pack-rolled Ti-46Al-8Nb alloy, leading to the scatter microstructure and mechanical properties. The multistructure above is mainly from the local flow softening process due to the original microstructural characteristics and solute segregation, and it can be seen that the design of TiAl alloy with the full and fine α2/γ lamellar colony, low Al segregation, and the improved hot-mechanical treatment should be required for the homogeneous, heat-deformed microstructure for TiAl sheet fabrication.

Based on the above observation, the schematic illustration of the deformation processes of Al-enriched ribbons (Al segregation regions) and the elongated lamellar colonies ribbons (regions with net-like Nb segregations) can be deduced, as shown in Figure 8. Figure 8a illustrates the morphology of the as-cast microstructure of Ti-46Al-8Nb alloy, which consists of nearly full lamellar colonies with a few of the γ and B2 grains at colony boundaries. The net-like Al and Nb-segregation exist at boundaries and in colonies, respectively, as indicated by the labels. During the initial deformation period, as shown in Figure 8b, the lamellae are severely elongated, bent, and kinked. Different deformed lamellar morphologies result from the different orientations of the primary lamellae. In Nb segregation regions, mainly in lamellar colonies, B2/β phase would precipitate during the hot pack rolling process. Soft β grains can assist in lamella deformations, such as the kinking of lamellar colonies (the right colony in Figure 8b) at rolling temperatures. Finer multiple-phase grains would be formed near the kinking points. At the same time, soft Al segregation regions are easily deformed because of stress concentration. Massive γ grains are present at boundaries which would further accelerate the grain rotation and crystal boundary migration leading to the local softening at these regions. With the progress processing and the large strain accumulation, segregations are almost eliminated. However, the microstructural morphologies resulting from the segregations would be reserved as shown in Figure 8c. The mixed ribbon structure consisted of residual lamellae, fine lamellar colonies, and multiple-phase grains in Nb segregation regions, and recrystallized γ grains in Al segregation regions can be seen. The typical as-rolled microstructure for Ti-46Al-8Nb alloy is thus generated.

## 4. Conclusions

In this study, the pack rolling process and the microstructural evolution for Ti-46Al-8Nb sheet fabrication were investigated. The results and conclusions can be summarized as follows:(1)The sandwich structure of stainless steel/TiAl matrix has good deformation compatibility due to the approximate high-temperature strength during the hot compression process, showing a less deformation-resistant and homogeneous deformation process, but a severe interfacial reaction is observed leading the aggravation of the contact interface. For the TC4/TiAl structure, the interfacial reaction can be neglected, but wavy reduction and high deformation resistance occur due to the low high-temperature strength of the titanium alloy package.(2)Considering the deformation compatibility and the interface protection, the pack-rolled Ti-46Al-8Nb alloy is complexly designed with an outer stainless steel/isolation layer; an inner TC4 alloy/coating structure for the sheet fabrication; and finally, the sound TiAl sheet with the intact interface. No crack defects can be observed with 85% reduction under the rolling conditions, showing the proper packing structure design and rolling conditions.(3)The pack-rolled TiAl sheet takes on the alternate structure of the dark grain-boundary Al-enriched ribbons and the elongated lamellar colonies ribbons, in which the multistructure of the deformation ribbons’ structure, lumpy γ phase, and retained lamellar colony is obtained mainly from the local flow softening process due to the original microstructure and solute segregation in pack-rolled Ti-46Al-8Nb alloy.

## Figures and Tables

**Figure 1 materials-13-00762-f001:**
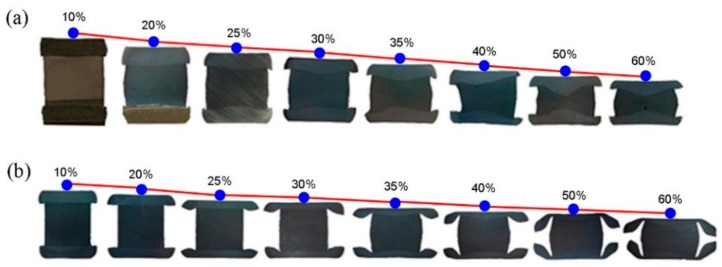
Macro-images of the sandwich samples of Ti-46Al-8Nb alloy after the hot compression process with the different packaging materials and deformation degrees: (**a**) stainless steel, (**b**) TC4 titanium alloy.

**Figure 2 materials-13-00762-f002:**
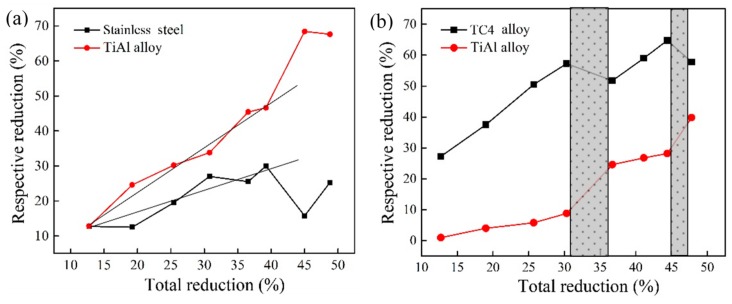
Respective reductions of the packaging materials and TiAl matrix during the compression process of the sandwich structure: (**a**) stainless steel package, (**b**) TC4 alloy package.

**Figure 3 materials-13-00762-f003:**
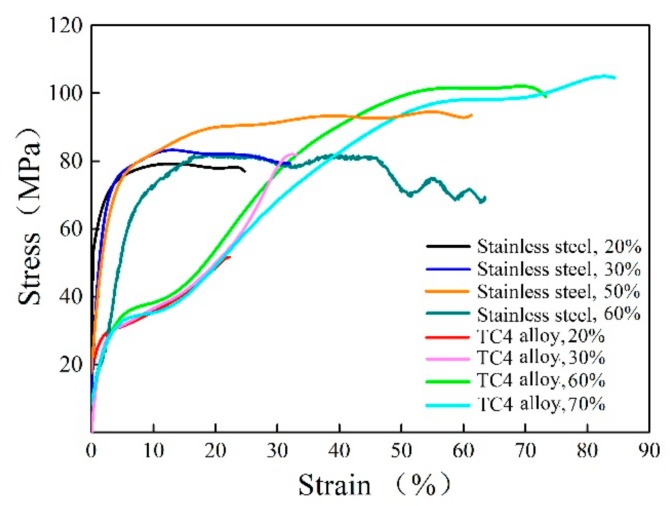
Stress-strain curve of the sandwich structure with the stainless steel andTC4 alloy package during the hot-compression process.

**Figure 4 materials-13-00762-f004:**
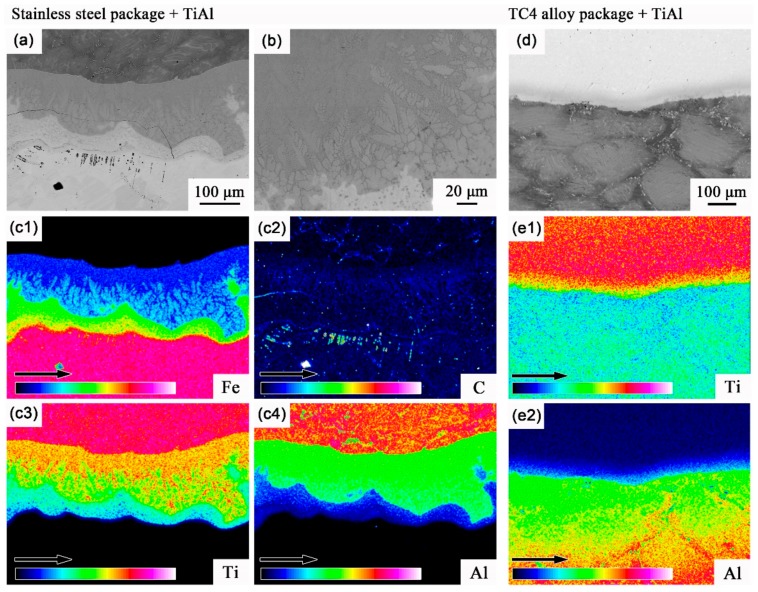
Morphology and solute distribution at the contact interface between the packaging layer and the Ti-46Al-8Nb matrix: (**a**) microstructure, (**b**) reaction interface, and (**c1**–**c4**) solute distribution for the stainless steel package; (**d**) microstructure and (**e1**,**e2**) solute distribution for the TC4 alloy package.

**Figure 5 materials-13-00762-f005:**
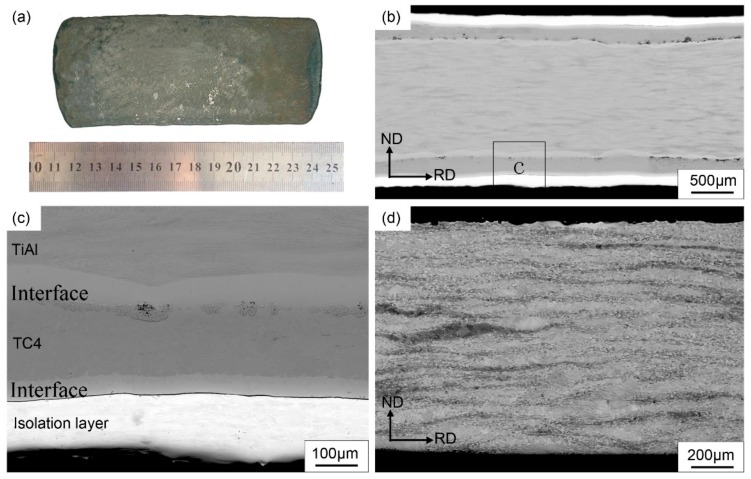
Macro and microstructural characteristics of the pack-rolled Ti-46Al-8Nb alloy with the complex package design: (**a**) macrograph of TiAl sheet, (**b**,**c**) the contact interface, and (**d**) microstructure of hot-rolling TiAl sheet.

**Figure 6 materials-13-00762-f006:**
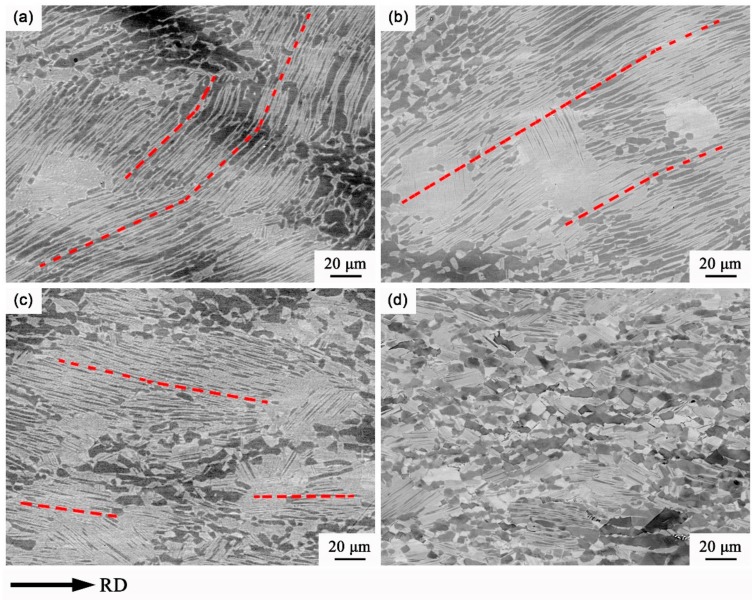
Microstructural evolution of the pack-rolled Ti-46Al-8Nb alloy with the different deformation degrees: (**a**) 30%, (**b**) 40%, (**c**) 60%, and (**d**) 80% (the horizontal direction is the rolling direction).

**Figure 7 materials-13-00762-f007:**
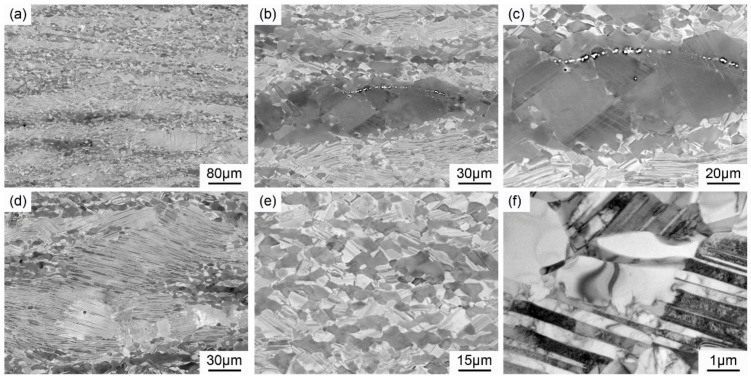
Multiple microstructures of the pack-rolled Ti-46Al-8Nb alloy with a reduction of 80%. (**a**) Deformed-ribbons structure, (**b**) local recrystallization, (**c**) lumpy γ phase at the grain boundary region, (**d**) retained lamellar colony, (**e**) fine recrystallized structure, and (**f**) TEM images for lamellar decomposition.

**Figure 8 materials-13-00762-f008:**
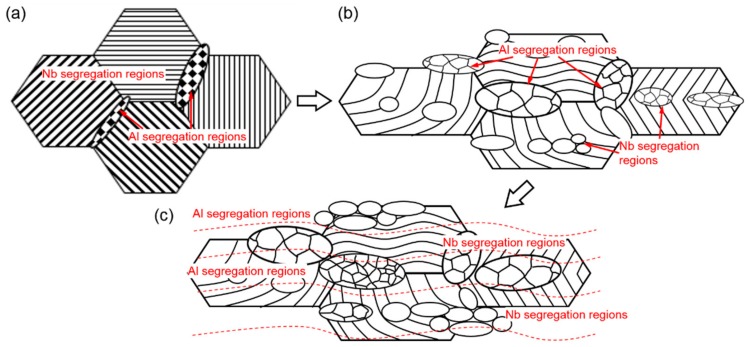
Schematic illustration showing the deformation processes during hot pack rolling for Ti-46Al-8Nb alloys. (**a**) As-cast microstructure, (**b**) the microstructure during the initial deformation period, and (**c**) the microstructure with large strain accumulation.

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
