# Peer review of "The Effects of Hot-Pack Coating Materials on the Pack Rolling Process and Microstructural Characteristics during Ti-46Al-8Nb Sheet Fabrication"

_materials, 2020, doi:10.3390/ma13030762_

Round 1

Reviewer 1 Report

The paper in hand deals with the pack rolling process and more particularly on the choice of an appropriate package material and the deformation of Ti-46Al-8Nb. While the results  seem coherent, they are poorly described by the authors and are hard to understand. I would suggest working on the text and improving the figure captions. Some comments are given below:

The most recent reference is from 2015 - consider including more recent work related to the topic. The authors chose to work on Ti-46Al-8Nb, however the alloy is not presented at all by the authors. Furthermore interfacial reactions that occur between sheet and alloy are not mentioned in the introduction. References for the instruments used are missing in the Materials and Methods section. Suppliers of sheet materials as well.  It is unclear whether the TiAl alloy used for section 3.1 is Ti-46Al-8Nb.  The authors labeled the sheets as stainless steel and titanium alloy in Figures 2 and 3. labelling the titanium alloy as TC4 would be more helpful as that is how it is refered to in the text. Figure 5 is very confusing. Consider adding the material to which each micrograph corresponds in the caption to help the reader find the information stated in the text easily. The authors keep mentioning the activity of the TiAl alloy. It is unclear to what it is refered until the middle of the paragraph. Consider using chemical affinity instead.  The 3rd paragraph in section 3.2 is very confusing. The text needs to be re-written in a clearer way.  Please define DRX before using the acronym

Overall interesting study but very hard to read.

Author Response

Response to Reviewer 1 Comments

The paper in hand deals with the pack rolling process and more particularly on the choice of an appropriate package material and the deformation of Ti-46Al-8Nb. While the results seem coherent, they are poorly described by the authors and are hard to understand. I would suggest working on the text and improving the figure captions. Some comments are given below:

-------------------------------------------------------------------------------------------------------

Points: The most recent reference is from 2015 - consider including more recent work related to the topic. The authors chose to work on Ti-46Al-8Nb, however the alloy is not presented at all by the authors. Furthermore interfacial reactions that occur between sheet and alloy are not mentioned in the introduction. References for the instruments used are missing in the Materials and Methods section. Suppliers of sheet materials as well. It is unclear whether the TiAl alloy used for section 3.1 is Ti-46Al-8Nb. The authors labeled the sheets as stainless steel and titanium alloy in Figures 2 and 3. labelling the titanium alloy as TC4 would be more helpful as that is how it is refered to in the text. Figure 5 is very confusing. Consider adding the material to which each micrograph corresponds in the caption to help the reader find the information stated in the text easily. The authors keep mentioning the activity of the TiAl alloy. It is unclear to what it is refered until the middle of the paragraph. Consider using chemical affinity instead. The 3rd paragraph in section 3.2 is very confusing. The text needs to be re-written in a clearer way. Please define DRX before using the acronym.

Overall interesting study but very hard to read.

-------------------------------------------------------------------------------------------------------

Response: Thanks for your thoughtful comments.

Some recent related works have been cited. The microstructure of Ti-46Al-8Nb, used for hot compression in this work, has been published in our previous works (DOI number: 10.1007/s40195-018-0742-4). To keep from repeating the data multiple times, we cited the article. We have replenished and rearranged the introduction. The instruments in the Materials and Methods section have been supplied in the manuscript, and the sheet materials of Ti-46Al-8Nb alloy is indicated for understanding the manuscript. The sheet material in Figures 2 and 3 has been labeled as TC4 alloy in the text. Figure 5 has been replenished for understanding the texteasily. We have used ‘chemical affinity’ in the text instead of ‘activity’. We have rewritten the 3rd paragraph in section 3.2. ‘DRX’ is the acronym of dynamic recrystallization which has been illustrated in the text. ‘The deformed microstructure of the local dynamic recrystallized (DRX) grains and elongated retained-lamellar colonies is obtained…’

Reviewer 2 Report

This article describes pack rolling processes in TiAl sheet fabrication using different package materials in hot compression experiments. Microstructure characteristics were studied. In general, the work is well presented with interesting results.

Firstly, the English used in the manuscript needs to be corrected and improved.

I would like the introduction to be expanded a little bit more for those readers who are not directly implicated in this field. For example, if there are a wide range of applications then these should be stated.

I would also like to see in the introduction concise aims and objectives of this study. This wasn’t clear to me.

Author Response

Response to Reviewer 2 Comments

This article describes pack rolling processes in TiAl sheet fabrication using different package materials in hot compression experiments. Microstructure characteristics were studied. In general, the work is well presented with interesting results.

-------------------------------------------------------------------------------------------------------

Point 1: The English used in the manuscript needs to be corrected and improved.

Response 1: Thanks for your comments. The English punctuation and grammar have been improved.

-------------------------------------------------------------------------------------------------------

Point 2: I would like the introduction to be expanded a little bit more for those readers who are not directly implicated in this field. For example, if there are a wide range of applications then these should be stated. I would also like to see in the introduction concise aims and objectives of this study. This wasn’t clear to me.

Response 2: Thanks for your suggestions. We have rewritten the introductionfor understanding the manuscript easily.TiAl sheets can be supplied for the skins, nozzles tiles and thermal protection systems (TPS) in aerospace vehicles with a wide range application,and the only success of the large-scale sheets (1800×500×1mm3) is achieved by GKSS and Plansee company with the advanced plate rolling process (ASRP) technology.

Reviewer 3 Report

The authors studied the optimal package structure design and the microstructure characteristics of the pack-rolling Ti46Al8Nb alloy as a candidate material for high temperature structural applications. In their work they considered different materials, such as stainless steel 0Cr25Ni20 and Ti6Al4V alloy, to develop their investigation.  

   The results obtained and conclusions of the study are worth of attention due to an improved pack structure design and rolling conditions presented. Before publishing, however, the paper needs text editing in sites, as indicated in the Enclosure. For instance, see line 193 for English correction. The authors should separate units from the numbers (e.g. 1.5 mm), but not separate <per cent> from "atomic", as "%" sign is not a unit. Besides, in References the authors should unify Journal titles notations in <italics>.  

Author Response

Response to Reviewer 3 Comments

The authors studied the optimal package structure design and the microstructure characteristics of the pack-rolling Ti46Al8Nb alloy as a candidate material for high temperature structural applications. In their work they considered different materials, such as stainless steel 0Cr25Ni20 and Ti6Al4V alloy, to develop their investigation.The results obtained and conclusions of the study are worth of attention due to an improved pack structure design and rolling conditions presented.

-------------------------------------------------------------------------------------------------------

Point: Before publishing, however, the paper needs text editing in sites, as indicated in the Enclosure. For instance, see line 193 for English correction. The authors should separate units from the numbers (e.g. 1.5 mm), but not separate <per cent> from "atomic", as "%" sign is not a unit. Besides, in References the authors should unify Journal titles notations in <italics>.

Response: Thanks for your serious attitude. We have revised the text as indicated in the enclosure, including the English mistakes, the units, and the journal title notations in References.

Reviewer 4 Report

The paper is interesting but requires some major modification as per below suggestions :

Please reformulate the introduction as is difficult to follow

Re arrange Figure 4 because as is now is difficult to follow (normally is a, b, c, d, not  a,b, d)

You stated that the samples were prepared with a  “ standard metallographic techniques”. Please describe it in details   

In Figure 3 I am wondering what happened for the curve of “ stainless steels 60%” as it have a different yield for elastic part compared with other 3 curves

Did you have performed only a single trial for each reduction ? How can be statistical representative

“Finally, the macrostructure of the high-deformed TiAl sheet takes on the alternate structure of the dark grain-boundary Al-riched ribbons and the elongated lamellar structure, as shown in Fig. 5(d)” Did you check these observation with EDS or against literature ?

In Figure 6 is not clear which is the rolling direction, please indicate clearly

“Furthermore, the hot-deformed …Fig 6(d), bring some citation for these observation as well as.

Which state of deformation corresponds Figure 7

You present a TEM images, but you don’t give any details in methods and materials about this techniques and samples preparation for TEM, please amend accordingly

The more operative slip systems… please cite this phrase because form your results is not evident

In Figure 8 you provides 3 caption, a,b, c but not any description was found in Figure 8. Schematic illustration showing the deformation processes during hot-pack rolling for Ti-46Al-8Nb alloys …as well as they have this modification because are submitted to different strain temperature ? that are necessary to be depicted in these Figures for clear understanding  

It is required a section with discussion to clarify the mechanism observe din this paper

Some minor English issues are noted

Author Response

Response to Reviewer 4 Comments

The paper is interesting but requires some major modification as per below suggestions:

-------------------------------------------------------------------------------------------------------

Point 1: Please reformulate the introduction as is difficult to follow

Response 1: Thank you. We have reformulated the introduction to make it easy to follow.

-------------------------------------------------------------------------------------------------------

Point 2: Rearrange Figure 4 because as is now is difficult to follow (normally is a, b, c, d, not a, b, d)

Response 2: Thanks for your suggestion. We have rearranged Fig.4.

-------------------------------------------------------------------------------------------------------

Point 3: You stated that the samples were prepared with a “standard metallographic techniques”. Please describe it in details

Response 3: Thanks for your suggestion. The standard metallographic techniques for preparing OM and SEM samples were supplemented in Materials and Methods section.

-------------------------------------------------------------------------------------------------------

Point 4: In Figure 3 I am wondering what happened for the curve of “ stainless steels 60%” as it have a different yield for elastic part compared with other 3 curves

Response 4: Thanks for your question. The “sandwich structure” hot compression experiments were directly conducted for the effect of the different package materials, and the hot deformation behaviours and the strain-stress curve depend on the deformation degree and the complex interface reaction for the package material and the as-cast TiAl ingots, which may lead to the different yield process during the compression process.

-------------------------------------------------------------------------------------------------------

Point 5: Did you have performed only a single trial for each reduction? How can be statistical representative

Response 5: Thanks for your suggestion. The “sandwich structure” hot compression can reproduce the rolling conditions to obtain the plentiful useful flow data for pack-rolling TiAl alloys. The compression samples were heated to 1250 ℃ at a rate of 10 ℃/s, and then the wide range of the reduction (10%~60%) was conducted at the strain rate of 1 s-1. Three trials were performed for each reduction to reduce errors and be more significant statistically.

-------------------------------------------------------------------------------------------------------

Point 6: “Finally, the macrostructure of the high-deformed TiAl sheet takes on the alternate structure of the dark grain-boundary Al-riched ribbons and the elongated lamellar structure, as shown in Fig. 5(d)” Did you check these observation with EDS or against literature ?

Response 6: Thanks for your comments. The morphology of the dark grain-boundary Al-riched ribbons has been reported in the studies of Li et al., and we have added the work as reference [25] in Results and Discussion section.

-------------------------------------------------------------------------------------------------------

Point 7: In Figure 6 is not clear which is the rolling direction, please indicate clearly

“Furthermore, the hot-deformed …Fig 6(d), bring some citation for these observation as well as.

Response 7: Thank you very much for your comments. In Fig. 6, the horizontal direction is the rolling direction which has been supplemented in the Fig. 6 caption. Moreover, some citations were added for the deformation microstructure descriptions.

-------------------------------------------------------------------------------------------------------

Point 8: Which state of deformation corresponds Figure 7

Response 8: Thanks for your question. The streamline structure, multiple- recrystallized grains and retained lamellar with rolling reduction of 80% are observed in pack-rolling Ti-46Al-8Nb alloy, as shown in Fig. 7.

-------------------------------------------------------------------------------------------------------

Point 9: You present a TEM images, but you don’t give any details in methods and materials about this techniques and samples preparation for TEM, please amend accordingly.

Response 9: Thanks for your comments. We have supplemented the TEM sample preparation method in the Materials and Methods section. Specimens for TEM were prepared by twin jet polishing. This was carried out in a solution consisting of 5% perchloric acid, 35% butanol and 60% methanol (in volume fraction) at −25 °C at a voltage of 25-30 V.

-------------------------------------------------------------------------------------------------------

Point 10: The more operative slip systems… please cite this phrase because form your results is not evident

Response 10: Thanks for your comments. References [29, 30] were added for the deformation mechanisms.

-------------------------------------------------------------------------------------------------------

Point 11: In Figure 8 you provides 3 caption, a, b, c but not any description was found in Figure 8. Schematic illustration showing the deformation processes during hot-pack rolling for Ti-46Al-8Nb alloys …as well as they have this modification because are submitted to different strain temperature? that are necessary to be depicted in these Figures for clear understanding

Response 11: Thanks for your comments. Fig. 8 shows the schematic illustration of the deformation processes during hot pack rolling of Al-riched ribbons (Al segregation regions) and the elongated lamellar colonies ribbons (regions with net-like Nb segregations). The microstructure evolutions were submitted to deformation rolling reductions as indicated by the descriptions of “During the initial deformation period…” and “With the progress processing and the large strain accumulation…”

-------------------------------------------------------------------------------------------------------

Point 12: It is required a section with discussion to clarify the mechanism observe din this paper. Some minor English issues are noted

Response 12: Thanks for your comments. In this study, the results and discussions were both contained in Section 3. The pack rolling process and microstructure characteristic for Ti-46Al-8Nb sheet fabrication were investigated. Firstly, different package materials were selected for the “sandwich structure” hot compression experiments. Then, the deformation compatibility, stress-strain conditions and surface reaction were studied between the package layer and TiAl matrix. The optimal package structure design for TiAl sheet fabrication was presented. Finally, the microstructure characteristic and flow behaviour were investigated for the pack-rolling Ti-46Al-8Nb alloy.

Reviewer 5 Report

The submitted manuscript entitled ‘Effects of hot-pack coating materials on pack-rolling process and microstructure characteristic during Ti4 46Al-8Nb sheet fabrication’ deals with the pack rolling of a hardly deformable Ti alloy at high temperature. The manuscript is interesting, but in its current form cannot be accepted by this Reviewer for publishing. During the review, a list of problems arose.

- Please provide official e-mail addresses for each Author, instead of commercial ones.

- The Abstract is messy, pleas rewrite completely.

- Linespacing is incorrect in Section 2.

- ‘The as-cast cylindrical Ti-46Al-8Nb ingot with an actual composition of Ti-46.3Al-7.8Nb (at. %) was fabricated by the induction skull melting(ISM).’ – how was the chemical composition measured? Are the Authors sure about the dimension (‘at%’)?

- ‘Subsequently the homogenization heat treatment was performed at 1200℃ for 4h. The microstructure above is comprised of the α2/γ lamellar colony, B2 phase and grain-boundary γ phase.’ – please insert micrograph(s).

- Please always let a space between the value and its unit, except in the case of ‘°C’ and ‘%’.

- ‘…during which the stainless steel (0Cr25Ni20) and titanium alloy (TC4, Ti-6Al-4V) were selected as the package material…’ – please add chemical compositions (the stainless steel designation seems to be incomplete) and heat treatment conditions.

- In the opinion of this reviewer, the statement of the sentences ‘For the selected sandwich structure, the metal plate is placed between two indenters where the material flow is promoted along its width and neglected along the length in the plane strain compression. The hot compression above can reproduce the rolling conditions to obtain the plentiful useful flow data for pack-rolling TiAl alloys.’ is false. Please add stress analysis to confirm the sentences.

- Page 2, line 85: put ‘-1’ to superscript.

- Page 3 line 94 – please use subscripts in the chemical compounds.

- Please identify the elements (plates, sheet) in fig 1.

- Can the waviness in the case of higher deformations (fig 1a, above 40%) be connected to improper preparation of the experiments?

-Are the curves in fig 3 average curves or individual ones? If the first, then please add scatter bands, if the latter, then repetition of the experiments is needed and suggested.

- Please add colour legends to fig 4. The labels of the subfigs are confused, please rearrange.

- Fig 6 is way too blurry for publishing, please replace. The red markers are invisible in the subfigs, please thicken and recolour.

Author Response

Response to Reviewer 5 Comments

The submitted manuscript entitled ‘Effects of hot-pack coating materials on pack-rolling process and microstructure characteristic during Ti-46Al-8Nb sheet fabrication’ deals with the pack rolling of a hardly deformable Ti alloy at high temperature. The manuscript is interesting, but in its current form cannot be accepted by this Reviewer for publishing. During the review, a list of problems arose.

-------------------------------------------------------------------------------------------------------

Point 1: Please provide official e-mail addresses for each Author, instead of commercial ones.

Response 1: Thanks for your comments. We have provided the official e-mail addresses instead of commercial ones.

-------------------------------------------------------------------------------------------------------

Point 2: The Abstract is messy, pleas rewrite completely.

Response 2: Thank you. We have rewritten the abstract to make it easy to follow.

-------------------------------------------------------------------------------------------------------

Point 3: Line spacing is incorrect in Section 2.

Response 3: Thank you. We have corrected the line spacing of Section 2.

-------------------------------------------------------------------------------------------------------

Point 4: ‘The as-cast cylindrical Ti-46Al-8Nb ingot with an actual composition of Ti-46.3Al-7.8Nb (at. %) was fabricated by the induction skull melting(ISM).’ – how was the chemical composition measured? Are the Authors sure about the dimension (‘at%’)?

Response 4: Thanks for your questions. Chemical analysis results were analyzed by Inductively Coupled Plasma Mass Spectrometry (ICPMS) for both the alloys, and the atomic percent was determined. The test method was supplemented in Materials and Method section.

-------------------------------------------------------------------------------------------------------

Point 5: ‘Subsequently the homogenization heat treatment was performed at 1200℃ for 4h. The microstructure above is comprised of the α2/γ lamellar colony, B2 phase and grain-boundary γ phase.’– please insert micrograph(s).

Response 5: Thanks for your comments. The microstructure, used for hot compression in this work, has been published in our previous works (DOI number: 10.1007/s40195-018-0742-4). To keep from repeating the data multiple times, we cited the article.

-------------------------------------------------------------------------------------------------------

Point 6: Please always let a space between the value and its unit, except in the case of ‘°C’ and ‘%’.

Response 6: Thank you. We have left a space between each value and its unit except in the case of ‘°C’ and ‘%’.

-------------------------------------------------------------------------------------------------------

Point 7: ‘…during which the stainless steel (0Cr25Ni20) and titanium alloy (TC4, Ti-6Al-4V) were selected as the package material…’ – please add chemical compositions (the stainless steel designation seems to be incomplete) and heat treatment conditions.

Response 7: Thanks for your comments. We have added the chemical compositions of 2520 stainless steel in Materials and Method section.

-------------------------------------------------------------------------------------------------------

Point 8: In the opinion of this reviewer, the statement of the sentences ‘For the selected sandwich structure, the metal plate is placed between two indenters where the material flow is promoted along its width and neglected along the length in the plane strain compression. The hot compression above can reproduce the rolling conditions to obtain the plentiful useful flow data for pack-rolling TiAl alloys.’ is false. Please add stress analysis to confirm the sentences.

Response 8:Thanks for your question. The hot compression process of the “sandwich structure” is designed for the pack-rolling process, considering the stress-strain evolution in the case of smaller deformationsand the interface reactionin the case of higher deformations. Additionally, the references of 24 and 25 indicate the plane strain conditions with the dimension conditions of the sandwich structure for the deformation process, which is conducted in the experiments.

-------------------------------------------------------------------------------------------------------

Point 9: Page 2, line 85: put ‘-1’ to superscript.

Response 9: Thank you. We have put the ‘-1’ to superscript in the text. ‘… was conducted at the strain rate of 1 s-1.’

-------------------------------------------------------------------------------------------------------

Point 10: Page 3 line 94 – please use subscripts in the chemical compounds.

Response 10: Thank you. We have corrected the chemical formula as ‘10 ml HF–10 ml HNO3–180 ml H2O.’ in the text.

-------------------------------------------------------------------------------------------------------

Point 11: Please identify the elements (plates, sheet) in fig 1.

Response 11: Thank you.

-------------------------------------------------------------------------------------------------------

Point 12: Can the waviness in the case of higher deformations (fig 1a, above 40%) be connected to improper preparation of the experiments?

Response 12: Thanks for your question. The hot compression process of the “sandwich structure” is designed for the pack-rolling process, considering the stress-strain evolution and the interface reaction. The waviness in the case of higher deformations is mainly due to the interface reaction of the dissimilar materials. Additionally, the references of 24 and 25 indicate the plane strain conditions with the dimension conditions of the sandwich structure for the deformation process, which is conducted in the experiments.

-------------------------------------------------------------------------------------------------------

Point 13: Are the curves in fig 3 average curves or individual ones? If the first, then please add scatter bands, if the latter, then repetition of the experiments is needed and suggested.

Response 13: Thanks for your questions. The “sandwich structure” hot compression samples were heated to 1250 ℃ at a rate of 10 ℃/s, and then the wide range of the reduction (10%~60%) was conducted at the strain rate of 1 s-1. Three trials were performed for each reduction to reduce errors which has been suggested in Materials and Method section.The continuous stress-strain curve is verified showing the discrepancy of the deformation process for the different package materials, and then the scatter band is not added in Fig. 3.

-------------------------------------------------------------------------------------------------------

Point 14: Please add colour legends to fig 4. The labels of the subfigs are confused, please rearrange.

Response 14: Thanks for your comments. We have added subfigure labels and rearranged Fig. 4.

-------------------------------------------------------------------------------------------------------

Point 15: Fig 6 is way too blurry for publishing, please replace. The red markers are invisible in the subfigs, please thicken and recolour.

Response 15: Thanks for your comments. We have revised Fig. 6.

Round 2

Reviewer 4 Report

The author responded to most of reviewer question, however some answers are not detailed enough.

Please give a detailed procedure of sample prepartion for “standard metallographic techniques”, because in the actual format is diffciult to reproduce the process

Author Response

Response to Reviewer 4 Comments

The author responded to most of reviewer question, however some answers are not detailed enough.

-------------------------------------------------------------------------------------------------------

Points: Please give a detailed procedure of sample prepartion for “standard metallographic techniques”, because in the actual format is diffciult to reproduce the process

-------------------------------------------------------------------------------------------------------

Response: Thanks for your comments. The “standard metallographic techniques” for preparing OM and SEM samples were rubbed by #100, #400, #1000, and #1500 sandpapers, mechanically polished by Cr2O3 polishing solution and etched with a solution of 10 ml HF+10 ml HNO3+180 ml H2O. The procedure has been supplemented in the Materials and Methods section.

Reviewer 5 Report

Thank you for all the changes and corrections.

Author Response

Response to Reviewer 5 Comments

-------------------------------------------------------------------------------------------------------

Points: Thank you for all the changes and corrections.

-------------------------------------------------------------------------------------------------------

Response: Thanks for your patient guidance.